# Pilot Testing of the “Turbidimeter”, a Simple, Universal Reader Intended to Complement and Enhance Bacterial Growth Detection in Manual Blood Culture Systems in Low-Resource Settings

**DOI:** 10.3390/diagnostics12030615

**Published:** 2022-03-01

**Authors:** Barbara Barbé, Ellen Corsmit, Jasper Jans, Kamalpreet Kaur, Roel Baets, Jan Jacobs, Liselotte Hardy

**Affiliations:** 1Clinical Sciences Department, Institute of Tropical Medicine, 2000 Antwerp, Belgium; bbarbe@itg.be (B.B.); ecorsmit@itg.be (E.C.); jjacobs@itg.be (J.J.); 2Center for Nano- and Biophotonics (NB-Photonics), Ghent University, 9000 Ghent, Belgium; jasper.jans@ugent.be (J.J.); kamalpreet.kaur@ugent.be (K.K.); roel.baets@ugent.be (R.B.); 3Photonics Research Group, Department of Information Technology, Ghent University-imec, 9000 Ghent, Belgium; 4Department of Microbiology, Immunology and Transplantation, KU Leuven, 3000 Leuven, Belgium

**Keywords:** blood culture, bloodstream infection, bacterial growth, turbidimetry, low-resource settings

## Abstract

Bloodstream infections and antimicrobial resistance are an increasing problem in low-income countries. There is a clear need for adapted diagnostic tools. To address this need, we developed a simple, universal reader prototype that detects bacterial growth in blood culture bottles. Our “turbidimeter” evaluates bacterial growth, based on the turbidity of the broth and the color change of the colorimetric CO_2_ indicator in commercially available blood culture bottles. A total of 60 measurements were performed using 10 relevant microbial species, spiked in horse blood, to compare the turbidimeter’s performance with that of an automatic reference system. The turbidimeter was able to detect growth in all but one of the spiked blood culture bottles. In the majority (7/10) of the species tested, time-to-detection of the turbidimeter was shown to be non-inferior to the reference automated time-to-detection. This was, however, only the case when both the turbidity and color change in the colorimetric CO_2_-indicator were used to evaluate growth. We could not demonstrate the non-inferiority of the turbidity measurement alone. Overall, the turbidimeter performed well, but we also identified some improvements that will be implemented in the next version of the prototype.

## 1. Introduction

Bloodstream infections (BSI) are an important cause of morbidity and mortality worldwide. Global estimates for sepsis, a possibly lethal outcome of BSI, are 49 million cases and 11 million deaths per year [1]. Data—although scarce and incomplete—demonstrate that low-income countries are hit hardest by antimicrobial resistance (AMR) [2,3,4,5]. Even though BSI misdiagnosis and mistreatment are important factors in the rise of AMR [6], the true burden of BSI in low-income countries remains largely unknown, due to a lack of reliable diagnostic and surveillance capacity in these settings [7,8,9,10].

BSI diagnosis relies on blood cultures, the inoculation and subsequent incubation of blood in blood culture bottles (BCBs). Automated blood culture systems that continuously monitor bacterial growth in the BCBs and give an alert when bacterial growth has been detected have become the reference standard in high-income countries. Growth assessment by these systems is based on CO_2_ production by growing microorganisms in BCBs, which growth is detected through a color change (BacT/ALERT^®^, bioMérieux, Marcy-l’Etoile, France), a fluorescent signal (BACTEC^TM^, Becton, Dickinson and Company (BD), Franklin Lakes, NJ, USA), or redox variations (VersaTREK^TM^, Thermo Fisher Scientific, Waltham, MA, USA) [11]. “Manual” blood culture systems, on the other hand, consist of BCBs that are incubated in a conventional incubator, wherein bacterial growth is detected via visual inspection [7]. During the visual inspection of bacterial growth, different parameters are assessed, such as the appearance of turbidity (“cloudiness”) caused by undissolved particles in the broth, the deposition of bacterial colonies as “puff balls” at the bottom of the BCB, or as pellicle formation at the liquid–air interface, and gas production [7]. Automated systems outperform manual systems in terms of time-to-detection (TTD) and growth detection [7,11,12,13,14,15,16], but they are expensive, require regular maintenance, and are not adapted to the environmental conditions commonly seen in low-resource settings (LRS) [17]. Therefore, many LRS laboratories still resort to manual blood culture systems. In addition, the use of manual blood culture systems is growing and is forecasted to reach roughly two-thirds of the global blood culture market by 2025 [18].

Improved, simple and affordable methods for bacterial growth detection in blood cultures are urgently needed in LRS. This was reinforced by the World Health Organization in numerous different working groups and guidelines [19,20]. Research is ongoing on novel techniques for growth detection, possibly combined with simultaneous identification [21,22,23], but these techniques have yet to demonstrate their performance and achieve a successful market introduction. To inform product developers, Dailey et al. [24] established a target product profile with requirements for a simplified blood culture system tailored to LRS. Even a system that meets part of these requirements and that takes into account the procurement and capacity challenges of LRS would already be a substantial improvement over the currently available systems. In addition, it would enable an expansion of the use of blood cultures in LRS and enhance BSI management in these settings [24].

To address this need, we developed a simple, universal reader prototype that detects bacterial growth in BCBs. This “turbidimeter” assesses blood culture broth turbidity in the BCB. Turbidimetry is fast and non-destructive [25] and is the most important parameter for growth when manually evaluating blood cultures [7]. An objective method to detect an increase in turbidity will shorten TTD in manual blood cultures. A second parameter for growth that was evaluated in this study was the color change of the CO_2_ indicator seen in commercial BacT/ALERT BCBs (bioMérieux). Here, we present the results of the “turbidimeter” pilot testing, in which the performance was tested in comparison with a reference method (automatic blood culture), along with the next steps in the development of this universal reader.

## 2. Materials and Methods

### 2.1. Turbidimeter Prototype: Concept and Design

The turbidimeter prototype was built using cheap off-the-shelf electronic components (e.g., discrete light-emitting diodes (LED), silicon photodetectors, and an Arduino development board) integrated in a custom 3D-printed holder that was designed with Fusion 360 (Autodesk Inc., San Rafael, CA, USA) and printed with the Original Prusa i3 MK3S+ 3D printer (Prusa Research, Prague, Czech Republic). The cost of equipment for one unit of the turbidimeter prototype was around USD 50. The holder used in this pilot testing was designed to take the polycarbonate “BacT/ALERT-type” bottles, but it can be customized to fit other BCB types of different materials or shapes.

The turbidimeter prototype evaluates bacterial growth based on two parameters. Light from a 598 nm LED is emitted through the blood culture bottle every 30 s and the transmitted light is captured by a silicon photodiode. As the turbidity of the broth increases with increased bacterial growth, the intensity of the transmitted light decreases. An additional red-green-blue (RGB) color sensor at the base of the turbidimeter is used to evaluate the color change of the colorimetric CO_2_ indicator on the bottom of the BacT/ALERT BCBs (Figure 1). Raw data are transferred in real time through a USB connection, in a CSV format.

### 2.2. Preparation of Spiked Blood Cultures

Defibrinated horse blood, spiked with known bacteria and yeast strains, was added to BacT/ALERT PF PLUS BCBs in concentrations and volumes relevant for pediatric blood cultures [26]. The strains from 10 clinically relevant species were selected for the growth experiments, comprising nine ATCC reference strains and one clinical isolate (Table 1). The strains were reactivated from −80 °C by sub-culturing twice on Columbia agar plates with 5% sheep blood (BD) and incubating overnight at 35–37 °C, after which colony counting was performed. Next, a 0.5 McFarland (~1.5 × 108 colony-forming units (cfu)/mL) bacterial suspension was prepared in sterile saline at 0.9%. For *Burkholderia cepacia*, a 0.75 McFarland suspension (~2.2 × 108 cfu/mL) was prepared. Just before initiating the growth experiments, the 0.5 (and 0.75) McFarland bacterial suspensions were diluted in sterile saline at 0.8% (twice, 1/100 dilution and 1/40 dilution), resulting in a concentration of approximately 375 cfu/mL or 550 cfu/mL (for *Burkholderia cepacia*). This concentration was verified by transferring 100 µL of the final dilution onto three blood agar plates. Briefly, 500 µL of the final dilution (corresponding to ~187 cfu) was transferred to 10 mL of defibrinated horse blood (the “mother dilution”) (E&O Laboratories Ltd., Bonnybridge, UK), resulting in a final concentration of ~10 cfu per ml of blood. The spiked horse blood was homogenized in a roller-mixer for 3 min. Next, 2 mL of the spiked horse blood was inoculated into the BCBs, then the inoculum was mixed by inverting the BCBs and the BCBs were incubated.

### 2.3. Growth Experiments with Spiked Blood Culture Bottles

Growth experiments consisted of three spiked BCBs with the same strain and one negative control (only horse blood), using four units of the turbidimeter prototype in parallel (Figure 2). Each growth experiment was performed in duplicate (two runs). The inoculated BCBs were placed in the turbidimeters inside a conventional incubator. The BCBs in the turbidimeters were incubated at a constant temperature of 35–37 °C, protected from light and without agitation, continuously for 20 h, 24 h, 36 h, or 96 h (depending on the spiked strain, see Table 1). The measurements of broth turbidity and the colorimetric indicator were performed in the turbidimeter inside the incubator every 30 s during the incubation time. One additional BCB was incubated using the automated reference system, the BacT/ALERT^®^ 3D 120 (bioMérieux, Marcy-l’Etoile, France), at 35 °C with continuous shaking. After the incubation period, the BCBs were subcultured on Columbia agar plates with 5% sheep blood (BD) to check the purity and growth of the isolates.

### 2.4. Definitions

Growth detection was defined as the percentage of spiked BCBs that showed confirmed growth (by subculture) of the total number of bottles (i.e., with and without growth). Time-to-detection (TTD) was defined as the delay between incubation and the detection of growth. TTD was automatically provided by the automated reference system and calculated from the growth curves of the turbidimeter measurements. The non-inferiority of the turbidimeter was evaluated by comparing the growth detection and TTD to the results from the reference system. For growth detection, a non-inferiority margin of 15% was determined; for TTD, the non-inferiority margin equalled 12 h.

### 2.5. Data Analysis

Data from the turbidimeter software were exported to Excel (Microsoft, Redmond, WA, USA) and contained the raw light intensity data as well as the colorimetric data measured by the RGB sensor. Raw light intensity data, as detected by the turbidimeter, were converted to turbidity by inverting the light intensity value, and turbidity was normalized based on the measurement data between 5 and 7 h after incubation. This was achieved using the formula “−log10(It/I0)” [27], where It is the light intensity measured at time t, and I0 is the average light intensity measured after between 5 and 7 h of incubation. RGB values were converted to hue saturation lightness (HSL) values [28]. Lightness was used as the parameter for the color change. An increase of 0.5% in lightness compared to the baseline was experimentally determined as the threshold for growth. The growth curves were analyzed visually, and the growth curve inflection point was calculated using the Growthcurver software package [29] for R (R Foundation for Statistical Computing, Vienna, Austria). The inflection point was used as a proxy for the TTD of the turbidimeter measurements. The graphs were created using R.

## 3. Results

### 3.1. Turbidimeter Prototype: Preparing for the Pilot Testing

The turbidimeter was extensively tested before the start of the pilot testing. In this testing phase, we stumbled upon a few limitations of the current prototype, which were considered and solved to the greatest extent possible before starting the study. A major limitation of our turbidity measurement is the disturbance of the signal because of the slow sedimentation of the blood, which also hinders shaking during incubation. Furthermore, the turbidimeter signal was seen to have outliers because of interference from electrical signals in the environment or because of the obstruction of the signal by undissolved particles.

#### 3.1.1. Sedimentation of Blood

The degree of sedimentation of the red blood cells in the BCBs, which is needed to assess the turbidity of the broth, took several hours (up to six hours). Figure 3 illustrates the sedimentation process. The slow sedimentation of the blood has a major impact on turbidity measurements. Firstly, since the sedimentation process during the first hours of incubation obscured the turbidity measurement, the data from these hours were not used for data analysis. To normalize the data, we used the period of measurement that was most stable: between five and seven hours. At that moment, the blood was fully sedimented, but growth had not yet started, so the BCB broth was clear. All data values were corrected using the average of the values taken between five and seven hours. In addition, the first six hours of measurement were removed from the graphs. Secondly, in our pilot study. it was not possible to shake the BCBs during incubation, which may possibly shorten TTD [30,31] but interferes with the need for sedimentation of the blood to measure the turbidity.

#### 3.1.2. Measurement Data Collection

At first, we programmed the turbidimeters to have 10-minute intervals between measurements, similar to what happens in blood culture automated systems. This resulted in messy growth curves, with a considerable number of outliers that obscured the interpretation of the curves (both visually and via the Growthcurver software). Therefore, it was decided to decrease the time between measurements to 30 s. Furthermore, the normalization (as discussed above) also improved the output growth curves.

### 3.2. Detection of Growth Using the Turbidimeter

#### 3.2.1. Detection of Turbidity

Turbidity (measured by a decrease in the intensity of the transmitted light) was detected in only four out of 10 strains. For these, we examined the turbidity growth curves in more detail to evaluate the variability between replicates of the same inoculum, incubated in multiple turbidimeter modules. In Table 2, we have listed four variables from the Growthcurver software [29], including the mean and standard deviation of the replicate measurements (intra-measurement variation) and between runs (inter-measurement variation). The two most important variables to consider for the future applicability of the tool are sigma and t_mid. Sigma reflects the goodness of fit, which is the residual standard error from the fitted nonlinear regression model. A low sigma value indicates a good fit of the logistic curve and implies that the variables given by the Growthcurver software are correct. In our measurements, the sigma value was constantly low (maximum 0.06), which is also reflected in the well-fitted graphs (Figure 4). T_mid, or the inflection point, is the time at which the population density reaches ½ of the carrying capacity and is therefore considered as the “moment of positivity” (time-to-detection). The variation of t_mid was low between replicates in one run (0.11–0.55) and between different runs (0.17–0.36).

#### 3.2.2. Detection of Color Change in the CO_2_ Indicator

In all but three of the turbidimeter measurements, the color change of the CO_2_ indicator was detected by the RGB sensor. The RGB sensor did not detect growth in one of six *Pseudomonas aeruginosa*-spiked BCBs, resulting in one false-negative bottle, since turbidity was not detected either.

#### 3.2.3. Turbidimeter Performance in General

To evaluate the turbidimeter’s performance, the detection of growth in the spiked BacT/ALERT BCBs was determined based on two parameters: turbidity and the change in the lightness of the CO_2_ indicator. The TTD of the turbidimeter was based on the earliest sign of growth in one of the two parameters (Table 3). Examples of growth curves from the turbidimeter are shown in Figure 5. In terms of growth detection, the turbidimeter was non-inferior to the reference method: 59 out of 60 (98.3%) spiked blood cultures showed growth. The TTD was non-inferior to that of the reference system for seven strains. The turbidimeter was not able to detect growth in the BCBs spiked with *Candida albicans*, *Haemophilus influenzae*, and *Neisseria subflava* within a 12-hour period after the detection by the reference system.

## 4. Discussion

The turbidimeter was able to detect growth in all but one spiked BCB. In the majority (7/10) of the species tested, the TTD of the turbidimeter was shown to be non-inferior, meaning that growth was detected within 12 h of the reference automated device’s TTD. However, this was only the case when two parameters were taken into account to evaluate growth. We could not demonstrate the non-inferiority of the turbidimeter (versus the reference automated device) when only evaluating the turbidity. The RGB sensor, which was initially only added as an accessory feature (therefore, not much research went into this feature), was shown to be instrumental in detecting growth in most of the species.

We were able to detect growth via turbidity for four important bacterial species that cause BSI in LRS. The turbidimeter TTD of *Klebsiella pneumoniae*, *Streptococcus pneumoniae*, *Salmonella* Typhimurium, and *Escherichia coli* was comparable to the reference automated device’s TTD. For *Salmonella* Typhimurium, detection by turbidity was more than two hours faster compared to the reference automated device. The four species listed above are major BSI-causing microorganisms in LRS. *Klebsiella pneumoniae*, *Salmonella enterica* serotypes, and *Escherichia coli* were found in 45% of all positive blood cultures in Boko Hospital, Benin [32]. In the Democratic Republic of the Congo, they were identified in up to 70% of all positive blood cultures [33]. Their importance was also confirmed in other LRS surveillance projects [34,35,36,37]. The failure to detect turbidity in six species needs to be studied more fully and compared with simultaneous manual evaluation since we have demonstrated before that turbidity was the most sensitive sign of growth for all bacterial groups except for *Staphylococcus/Enterococcus* species [26]. However, these findings were from tests conducted with another type of commercially available BCB, which might have influenced the results.

Evaluation of the color change of the CO_2_ indicator of the commercially available BacT/ALERT BCBs was demonstrated to be very valuable for improving growth detection and the TTD of the system. Since this feature was only added as an accessory and not as the main “growth detector”, it might be possible to improve the TTD with this variable. However, the need for this variable also implies that the turbidimeter is less versatile than we initially had in mind. It could mean that the turbidimeter can only be used with BCBs that have a colorimetric CO_2_ indicator, limiting the use of the universal reader.

In a previous study, we evaluated the TTD of manual cultures when taken out of the incubator and checked for growth twice daily, as would be performed in a field laboratory that was open during office hours [26]. The TTD was at least 24 h for *Escherichia coli*, *Salmonella* Typhimurium, *Klebsiella pneumoniae*, *Streptococcus pneumoniae*, *Staphylococcus aureus*, *Burkholderia cepacia*, *Pseudomonas aeruginosa*; it took up to 48 h for *Candida albicans*, and up to 72 h for *Hemophilus influenzae* and *Neisseria subflava*. For the detection of growth, turbidity is the most used/useful variable, but it is very subjective, and a trained eye is needed to see subtle differences [7]. The turbidimeter has not been developed to serve as a stand-alone device but as an objective method to complement the visual inspection of manual blood cultures and speed up the TTD. In this study, continuous measurement (every 30 s) of the turbidity and the colorimetric CO_2_-indicator was performed to evaluate the complete growth curves and to identify thresholds for growth. When implementing the turbidimeter in the future, continuous measurement will not be needed: the turbidimeter can be used in an on/off mode and will give an extra parameter for growth during the visual inspection of manual blood cultures. One turbidimeter can be used to measure all BCBs.

There are limitations to our pilot study. We did not check for visual signs of growth because we did not want to interrupt the continuous measurement process. Therefore, we cannot properly evaluate the added value of the turbidimeter to manual blood cultures, we can only compare its performance with the reference automated device. We did use static blood cultures, unlike the continuous shaking in the reference automated device. This might have affected the growth of certain bacterial species. Since this was a pilot study, we only evaluated a limited number of species and replicates. Further studies will be performed with a larger set of species. In real settings, there will be problems with sampling (over- and under-filling), patients might have taken antibiotics before sampling, or there might be power cuts. The conditions in our laboratory were ideal (stable temperature and electricity, dust-free) and so do not mirror field conditions. A comparative field trial is planned in the very near future, in which the effect of environmental conditions (humidity, dust, temperature changes) will also be evaluated.

## 5. Conclusions

We have developed a method to complement the manual analysis of blood cultures in LR, improve the TTD, and thereby, the time until the correct treatment is administered. The device is portable, cheap, and easy-to-use, and is better adapted for use in LRS, compared to the available blood culture automated devices. In this pilot study, we demonstrated that the turbidimeter performed well using a small set of species, in ideal laboratory conditions, and with continuous measurement, but we also identified improvements to be made. The next turbidimeter prototype will include refined and additional features. In the coming months, this new prototype will be tested more exhaustively in our laboratory and afterward during field-testing in Western Africa.

## Figures and Tables

**Figure 1 diagnostics-12-00615-f001:**
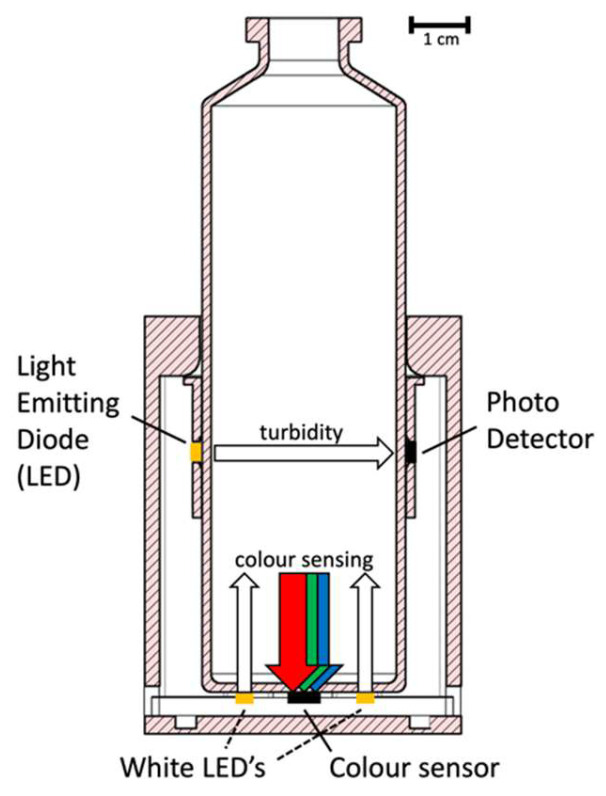
Light paths of the turbidimeter. Turbidity is measured horizontally through the blood culture bottle: the 598 nm (amber) LED is pulsed every 30 s, then the light intensity is measured by the photodetector opposite the LED, converted to digital data, and transferred in real time through the USB connection. Color is measured vertically at the bottom of the blood culture bottle (where the colorimetric CO_2_-indicator is located): two white LEDs are pulsed every 30 s, the color of the CO_2_-indicator is measured by the RGB sensor and then transferred in real time through a USB connection.

**Figure 2 diagnostics-12-00615-f002:**
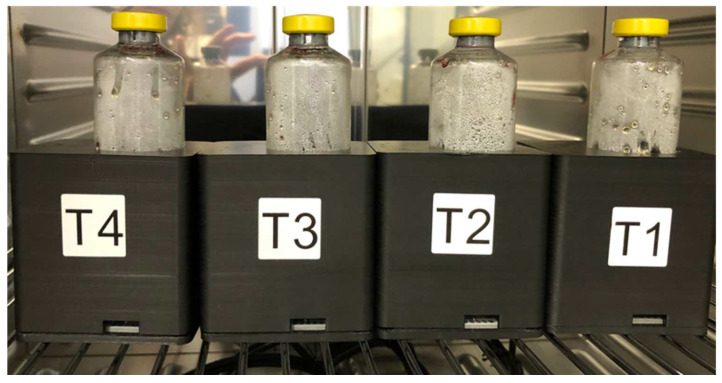
Setup of one growth experiment (1 run): three units of the turbidimeter prototype (T2-T3-T4) with BCBs containing spiked blood from the same “mother dilution” and one unit (T1) with a blank BCB (negative control, blood only). The BCBs in the turbidimeters are incubated in a dedicated incubator that remains closed during the growth experiment. Measurement data is then transferred to a laptop outside the incubator in real time, over a USB connection.

**Figure 3 diagnostics-12-00615-f003:**
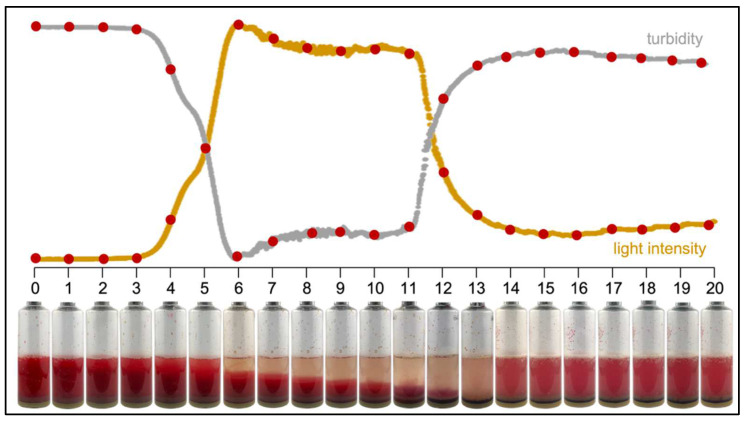
Illustrative time-lapse of a blood culture bottle spiked with *Escherichia coli* and incubated at 35–37 °C (one picture taken each hour). The graph depicts the intensity of the light of the LED shining through the BCB broth (yellow line) and the turbidity of the broth (grey line) which is the inversion of the light intensity signal (here shown without normalization). The sedimentation of blood takes up to 6 h, followed by a stable period in which the BCB broth remains clear until turbidity starts increasing after 11 h. This figure is a composite image: pictures were taken from several experiments and combined into one time-lapse; the light/turbidity curve was added, for illustration only, from a non-interrupted experiment in the incubator.

**Figure 4 diagnostics-12-00615-f004:**
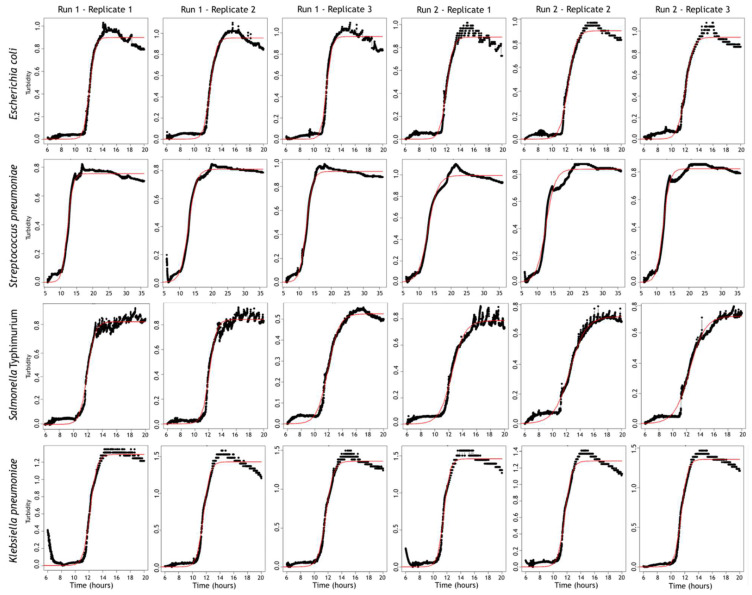
An overview of all turbidity growth curves, fitted by the Growthcurver software. The black lines are the turbidity values, fitted against the theoretical growth curves depicted by the thin red lines.

**Figure 5 diagnostics-12-00615-f005:**
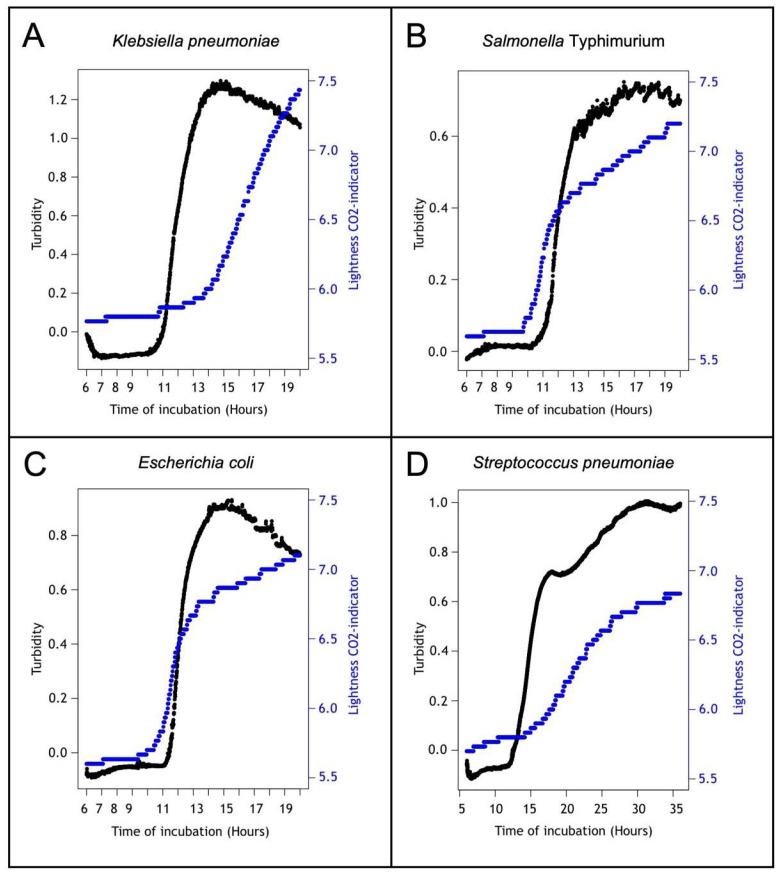
Example of growth curves over time for four strains. The black line (left *y*-axis) is the turbidity measured by the turbidimeter, while the blue line (right *y*-axis) is the shift in the lightness of the color indicator at the bottom of the BacT/ALERT blood culture bottle. Continuous measurement was performed for 20 h for *Klebsiella pneumoniae*, *Salmonella* Typhimurium, *Escherichia coli*, and, for 36 h, for *Streptococcus pneumoniae*. The growth curves shown here start from 6 h of incubation onward, as this is the time needed for the blood cells to settle in the BCB (which disturbs the light signal).

**Table 1 diagnostics-12-00615-t001:** Overview of the growth experiments: 2 runs with 3 replicates were performed for each strain.

N°	Strain	Mc Farland (McF) Suspension	Incubation Time
1	*Haemophilus influenzae* ATCC 49247	0.5 McF	96 h
2	*Escherichia coli* ATCC 25922	0.5 McF	20 h
3	*Staphylococcus aureus* ATCC 25923	0.5 McF	36 h
4	*Streptococcus pneumoniae* ATCC 49619	0.5 McF	36 h
5	*Burkholderia cepacia* ATCC 25416	0.75 McF	96 h
6	*Salmonella* Typhimurium ATCC 14028	0.5 McF	20 h
7	*Pseudomonas aeruginosa* ATCC 27853	0.5 McF	24 h
8	*Candida albicans* ATCC 66027	0.5 McF	96 h
9	*Klebsiella pneumoniae* ATCC 700603	0.5 McF	20 h
10	*Neisseria subflava* MO00045/1 (clinical isolate)	0.5 McF	96 h

**Table 2 diagnostics-12-00615-t002:** Details on turbidity measurement data for the four turbidity-positive species.

Strain	Run	Rep *	Growth Rate r	Mean ± SD r		Inflection Point t_mid	Mean ± SD t_mid		Doubling Time t_gen	Mean ± SD t_gen		Goodness- of-Fit Sigma	Mean ± SD Sigma	
*Escherichia coli*	1	1	2.71	2.59 ± 0.24	2.27 ± 0.41	12.09	12.09 ± 0.20	12.09 ± 0.17	0.26	0.27 ± 0.03	0.31 ± 0.06	0.05	0.05 ± 0.01	0.05 ± 0.01
		2	2.31			12.28			0.30			0.05		
		3	2.76			11.88			0.25			0.06		
	2	1	2.21	1.95 ± 0.23		11.97	12.10 ± 0.18		0.31	0.36 ± 0.04		0.05	0.05 ± 0.01	
		2	1.78			12.30			0.39			0.04		
		3	1.85			12.02			0.37			0.05		
*Streptococcus pneumoniae*	1	1	1.26	1.04 ± 0.21	0.92 ± 0.22	12.23	12.45 ± 0.34	12.50 ± 0.36	0.55	0.69 ± 0.14	0.79 ± 0.19	0.03	0.03 ± 0.00	0.03± 0.01
		2	0.84			12.84			0.83			0.02		
		3	1.02			12.27			0.68			0.03		
	2	1	0.67	0.80 ± 0.19		12.93	12.54 ± 0.46		1.03	0.90 ± 0.19		0.04	0.04 ± 0.01	
		2	0.70			12.66			0.99			0.04		
		3	1.01			12.01			0.68			0.03		
*Salmonella* Typhimurium	1	1	1.98	1.75 ± 0.42	1.36 ± 0.52	11.93	12.00 ± 0.11	12.26 ± 0.29	0.35	0.41 ± 0.12	0.57 ± 0.20	0.04	0.03 ± 0.01	0.03 ± 0.01
		2	2.01			12.13			0.34			0.04		
		3	1.27			11.94			0.55			0.02		
	2	1	1.11	0.96 ± 0.13		12.51	12.51 ± 0.03		0.62	0.73 ± 0.10		0.04	0.03 ± 0.01	
		2	0.86			12.54			0.80			0.03		
		3	0.91			12.49			0.76			0.03		
*Klebsiella pneumoniae*	1	1	2.14	2.08 ± 0.18	2.16 ± 0.17	12.19	11.85 ± 0.30	11.73 ± 0.24	0.32	0.34 ± 0.03	0.32 ± 0.03	0.07	0.06 ± 0.01	0.06 ± 0.01
		2	2.22			11.64			0.31			0.07		
		3	1.88			11.72			0.37			0.05		
	2	1	2.37	2.24 ± 0.13		11.58	11.62 ± 0.11		0.29	0.31 ± 0.02		0.07	0.06 ± 0.01	
		2	2.10			11.74			0.33			0.07		
		3	2.23			11.53			0.31			0.05		

***** Per run, three replicates were measured using different units of the turbidimeter prototype. SD is the variation between different units of the turbidimeter prototype, containing BCB with spiked blood from the same “mother dilution”.

**Table 3 diagnostics-12-00615-t003:** Median time-to-detection (TTD) of the turbidimeter and a comparison with the reference system.

Strain	TTD * Turbidity	TTD Colour Change	TTD Turbidimeter	TTD Reference System	TTD Turbidimeter Minus Reference System
*Escherichia coli*	12.1	11.4	11.4	12.0	−0.6
*Streptococcus pneumoniae*	12.5	17.4	12.5	11.6	0.9
*Salmonella* Typhimurium	12.3	10.9	10.9	13.2	2.3
*Klebsiella pneumoniae*	11.7	14.7	11.7	12.0	−0.3
*Staphylococcus aureus*	No detection	21.9	21.9	13.8	8.2
*Burkholderia cepacia*	No detection	27.3	27.3	20.5	6.8
*Pseudomonas aeruginosa*	No detection	18.7	18.7	16.7	2.0
*Haemophilus influenzae*	No detection	39.1	39.1	14.6	24.5
*Neisseria subflava*	No detection	57.4	57.4	17.0	40.4
*Candida albicans*	No detection	45.4	45.4	32.0	13.4

* TTD = time-to-detection in hours, the median of 3 replicates of 2 runs (6 measurements).

## Data Availability

The dataset is available from the figshare database: https://doi.org/10.6084/m9.figshare.19196741.v1 (accessed on 30 January 2022).

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
