# Peer review of "Pilot Testing of the “Turbidimeter”, a Simple, Universal Reader Intended to Complement and Enhance Bacterial Growth Detection in Manual Blood Culture Systems in Low-Resource Settings"

_diagnostics, 2022, doi:10.3390/diagnostics12030615_

Round 1

Reviewer 1 Report

This paper presents a simple and easy-to-operate device for the detection of turbidity and CO2 generated in BCB. The optimized protocol for BCBs proposed in this paper shortens the time required for effective positive detection of common clinical pathogens, which may have important implications for the prevention and treatment of bloodstream infections in low-income countries. I think this manuscript is acceptable after addressing the following issues

Q1: In paragraph 2.3, line 7, "Measurements of broth turbidity and colorimetric indicators were performed every 30 seconds during the incubation period". To achieve this frequency of testing, clinical practice must prepare a corresponding set of equipment for each blood culture. In a clinical application scenario, an adult patient needs to collect two to three sets of spiked blood samples, each of which needs to be collected from different locations to exclude contamination, which requires at least 6 to 9 corresponding BCBs devices. Therefore, this will greatly increase the cost of the hospital. So how can this device be used clinically?

Q2: Using silicon photodetectors instead of visual recognition increases the sensitivity of the system and reduces operator expertise, but environmental factors, especially unstable temperature and humidity in resource-poor environments, tend to interfere with sensors. This interference can lead to increased false-positive or false-negative rates of test results. How to prove the stability of the system working in different environments?

Q3: I noticed that the culture flasks used in this study are commercial products with CO2 indicators, and in resource-poor settings, blood culture flasks have the potential to be reused by autoclaving. So how to ensure that the culture flasks can be reused?

Author Response

First, we would like to thank all four reviewers for their constructive feedback. We have considered all suggestions, and strongly believe that we can now resubmit an improved manuscript. Below, detailed answers on all questions are given; the line numbers referred to are the line numbers in the track changes version.f

Reviewer 1

This paper presents a simple and easy-to-operate device for the detection of turbidity and CO2 generated in BCB. The optimized protocol for BCBs proposed in this paper shortens the time required for effective positive detection of common clinical pathogens, which may have important implications for the prevention and treatment of bloodstream infections in low-income countries. I think this manuscript is acceptable after addressing the following issues

Q1: In paragraph 2.3, line 7, “Measurements of broth turbidity and colorimetric indicators were performed every 30 seconds during the incubation period”. To achieve this frequency of testing, clinical practice must prepare a corresponding set of equipment for each blood culture. In a clinical application scenario, an adult patient needs to collect two to three sets of spiked blood samples, each of which needs to be collected from different locations to exclude contamination, which requires at least 6 to 9 corresponding BCBs devices. Therefore, this will greatly increase the cost of the hospital. So how can this device be used clinically?

Thank you for your concern. The answer to this comment is twofold. First, the cost of one turbidimeter prototype is currently 50 USD (this information was added on line 99) which is very low compared to the blood culture automates (+- 26.000 euro for a an automate with 120 cells, so >200 euro per cell). But more importantly, the turbidimeter is developed to be used in an on/off mode and will be an additional parameter to consider when working with manual blood cultures in low-resource settings. Only one unit is needed to evaluate all blood cultures incubated. Information on this was added on line 400-405.

Q2: Using silicon photodetectors instead of visual recognition increases the sensitivity of the system and reduces operator expertise, but environmental factors, especially unstable temperature and humidity in resource-poor environments, tend to interfere with sensors. This interference can lead to increased false-positive or false-negative rates of test results. How to prove the stability of the system working in different environments?

The stability of the system is indeed one of our concerns. Therefore, we will test the influence of environmental conditions in the field trial that will be done with the updated prototype (starting in September 2022 in Benin and Burkina Faso). We added more explanation on this on line 426-427.

Q3: I noticed that the culture flasks used in this study are commercial products with CO2 indicators, and in resource-poor settings, blood culture flasks have the potential to be reused by autoclaving. So how to ensure that the culture flasks can be reused?

The type of bottle that we have evaluated and are working with in our field sites is not reusable. Also, in low-resource settings, we prefer to work with plastic (not reusable) bottles over glass (reusable) bottles because plastic bottles offer better biosafety (low risk of breaking), lower weight (transport cost) and can be processed in a field incinerator), although they are less economical and cannot be recycled.

Reviewer 2 Report

Manuscript ID: diagnostics-1600375

Title:  Pilot testing of the “Turbidimeter”, a simple, universal reader intended to complement and enhance bacterial growth detection in manual blood culture systems in low-resource settings

General comments: This research paper by Barbé et al. demonstrated a “turbidimeter” that detects bloodstream infections and antimicrobial resistance based on the turbidity and color changes of the blood. The aim of this system is to implement the technology in low-resource settings in near future. Overall, the paper seems interesting. However, the main issue is optimization. I recommend author have to optimize the system well and move forward with increasing number of species. Author must improve the manuscript, especially optimization. The specific comments are listed below to improve the quality of this manuscript. Overall, major revision or rejection is recommended.

Comments:  

  1. Author must improve figure 1 for better understanding. Otherwise, the author should give detailed information of the turbidimeter’s working principle. Seems there is two white LED and RGB sensor for the measurement of color and turbidity, and there is no detailed information about the working principle and data collection in the figure or description.
  2. Did the author measure the temperature stability (even though it's white LED) of the turbidimeter? Was the temperature being stable throughout the experiment? I suggest giving the information if you have.
  3. Is any specific reason for choosing horse blood for the experiment and to spike? Why author did not choose the human blood which is a good simulation to establish the study to the field.
  4. Why the measurement was collected every 30 seconds? Did the author use an ADC to convert the light intensity to a digital signal? I recommend giving clear information to the readers
  5. why the turbidity was detected in only four out of 10 strains. Did the author fail to perform the assay or did the author face any challenge in bacterial growth?
  6. Author should provide the accuracy of the turbidimeter since you obtained the false negative in turbidity measurement for P. aeruginosa.
  7. Author should follow a consistent format to represent the species name as there is inconsistency throughout the manuscript. In addition, I highly recommend language editing.
  8. I strongly recommend performing a comparison study. Comparison between the standard method and your “turbidimeter”. The comparative analysis will reveal how sensitive and accurate your system is.
  9. Author intends to increase the number of species. Even though the number of species is one of the important parameters. Precise optimization of the system is the most important. Two or three species are enough to optimize the system that will eradicate the variabilities and generate consistent results.
  10. I recommend paraphrasing or removing the last two sentences in the abstract
  11. Introduction does not require any sub-heading. I suggest removing the sub-heading in the introduction.

Author Response

First, we would like to thank all four reviewers for their constructive feedback. We have considered all suggestions, and strongly believe that we can now resubmit an improved manuscript. Below, detailed answers on all questions are given; the line numbers referred to are the line numbers in the track changes version.

Reviewer 2

General comments: This research paper by Barbé et al. demonstrated a “turbidimeter” that detects bloodstream infections and antimicrobial resistance based on the turbidity and color changes of the blood. The aim of this system is to implement the technology in low-resource settings in near future. Overall, the paper seems interesting. However, the main issue is optimization. I recommend author have to optimize the system well and move forward with increasing number of species. Author must improve the manuscript, especially optimization. The specific comments are listed below to improve the quality of this manuscript. Overall, major revision or rejection is recommended.

Comments:  

  1. Author must improve figure 1 for better understanding. Otherwise, the author should give detailed information of the turbidimeter’s working principle. Seems there is two white LED and RGB sensor for the measurement of color and turbidity, and there is no detailed information about the working principle and data collection in the figure or description.

We have updated the caption of figure 1 to give more detailed information on the working principle (line 151-161).

  1. Did the author measure the temperature stability (even though it's white LED) of the turbidimeter? Was the temperature being stable throughout the experiment? I suggest giving the information if you have.

Unfortunately, we do not have information on the internal temperature of the turbidimeter since there was no temperature sensor included in this prototype. But we have added a temperature sensor in the new prototype. In future studies, we will be able to provide this information.

  1. Is any specific reason for choosing horse blood for the experiment and to spike? Why author did not choose the human blood which is a good simulation to establish the study to the field.

In Belgium nowadays, it is very difficult to purchase large volumes of human blood through the blood banks. We have encountered this problem in a previous project and have therefore decided to do a comparative study on bacterial growth in human versus horse blood. The results of this study are submitted to Plos Protocols and is currently in review. We can confirm that we obtained equivalent results in both types of blood.

  1. Why the measurement was collected every 30 seconds? Did the author use an ADC to convert the light intensity to a digital signal? I recommend giving clear information to the readers

The reason for the 30 seconds-interval was already explained in 3.1.2 Measurement data collection (lines 285-291). We chose this interval to have a large number of data points, which results in smoother curves. Information on the read out of the light intensity (indeed, an ADC was used to convert the light intensity in a digital signal) was added in the caption of figure 1 (line 151-161).

  1. why the turbidity was detected in only four out of 10 strains. Did the author fail to perform the assay or did the author face any challenge in bacterial growth?

This was surprising for us as well, since we had seen in a previous study that turbidity was the most sensitive sign of growth for all bacterial groups (except Staphylococcus/Enterococcus species). Growth was checked by subculturing after incubation and was always seen, so there was no problem with bacterial growth. In our future study (starting this month), using the updated prototype, we will also evaluate visually the sign of growth. This was not possible since we chose not to interrupt the incubation in the presented study. We have added a paragraph on this in the discussion (line 379-383).

  1. Author should provide the accuracy of the turbidimeter since you obtained the false negative in turbidity measurement for P. aeruginosa.

Thank you for your suggestion. The accuracy data will be evaluated in the next study, using the improved prototype (and combined with the visual signs of growth), on a larger set of samples.

  1. Author should follow a consistent format to represent the species name as there is inconsistency throughout the manuscript. In addition, I highly recommend language editing.

We have now written all genera + species without abbreviations, to avoid confusion. If the reviewer is referring to the way Salmonella Typhimurium is written, we would like to clarify this. Indeed, the full name is Salmonella enterica subsp. enterica, serovar Typhimurium. But it is generally accepted to abbreviate this name to Salmonella Typhimurium; Typhimurium is a serovar, not a species and should therefore not be written in italics. In addition, we have edited the language as well, as asked by the reviewer.

  1. I strongly recommend performing a comparison study. Comparison between the standard method and your “turbidimeter”. The comparative analysis will reveal how sensitive and accurate your system is.

This was a (pilot) comparison study, this is now more detailed in the description (lines 89-90). Performance of the turbidimeter was compared with a reference, the bioMérieux blood culture automate. Further comparison with manual + automatic blood cultures is being planned in the coming months with the new prototype.

  1. Author intends to increase the number of species. Even though the number of species is one of the important parameters. Precise optimization of the system is the most important. Two or three species are enough to optimize the system that will eradicate the variabilities and generate consistent results.

Thank you for your suggestion. But we believe, since different species have different growth characteristics, that we will need to include a larger number of species as well.  

  1. I recommend paraphrasing or removing the last two sentences in the abstract

The last two sentences were removed and the last sentence of the abstract was paraphrased (line 23).

  1. Introduction does not require any sub-heading. I suggest removing the sub-heading in the introduction.

We have deleted the subheadings in the introduction, as suggested.

Reviewer 3 Report

The manuscript „Pilot testing of the “Turbidimeter”, a simple, universal reader intended to complement and enhance bacterial growth detection in manual blood culture systems in low-resource settings” is interesting and presents the results obtained with a prototyped turbidimeter.

For the readers' convenience, the authors should consider the following recommendations:

Introduction-  present with more detail the existing diagnostic tools.

Materials and Methods - please, specify if the „four turbidimeters” refer to the prototype.

How many replicates were done with the same turbidimeter?

The values in Table 2 were obtained with the same turbidimeter or with the four of them? It is important to specify for understanding what mean±SD refers to.

Figures 3 and 4 - increase the font size of the axis titles.

Conclusion - highlight the advantages of this turbidimeter.

Author Response

First, we would like to thank all four reviewers for their constructive feedback. We have considered all suggestions, and strongly believe that we can now resubmit an improved manuscript. Below, detailed answers on all questions are given; the line numbers referred to are the line numbers in the track changes version.

Reviewer 3

The manuscript „Pilot testing of the “Turbidimeter”, a simple,  universal reader intended to complement and enhance bacterial growth  detection in manual blood culture systems in low-resource settings” is  interesting and presents the results obtained with a prototyped  turbidimeter.

For the readers' convenience, the authors should consider the following recommendations:

Introduction-  present with more detail the existing diagnostic tools.

Thank you for the suggestion. However, we believe that giving even more details on the three existing manufacturers is out of the scope of this manuscript, since this already spans 7 lines (lines 37-43).

Materials and Methods - please, specify if the „four turbidimeters” refer to the prototype.

How many replicates were done with the same turbidimeter?

The values in Table 2 were obtained with the same turbidimeter or  with the four of them? It is important to specify for understanding what  mean±SD refers to.

The three comments above were addressed at once. We apologize for the unclarity about this and have therefore added a picture of the turbidimeter modules with more information on the set-up of growth experiment in the caption (lines 208-213). Throughout the text on “preparation of spiked blood cultures” and “growth experiments with spiked blood culture bottles” (lines 166-197), we have added clarifications. In addition, we have added a footnote to table 2 to improve understanding of the standard deviation.

Figures 3 and 4 - increase the font size of the axis titles.

This was done, we reworked the figures for clarity.

Conclusion - highlight the advantages of this turbidimeter.

The conclusion was rephrased (lines 430-432) to highlight the advantages.

Reviewer 4 Report

This paper is very interesting and provides useful additional data for bacterial detection prototypes. In general, the paper is well written. “Materials and Methods” are easy to follow and also provide a complete and clear description of the device. “Results” are well presented, tables and figures are adequate. Lastly, “Discussion” is properly argued. 

However, additional modifications can improve the work. 

Major modifications:

  1. The introduction could be shortened as some phrases are more suitable in “Discussion”. 

  2. More importantly, the references are missing in the Introduction part of the manuscript. 

  3. Another concern of mine is the fact that some references that I decided to read (such as reference 9), do not have anything in common with the text, but are cited. These unintentional and accidental mistakes could have happened while writing, so I would recommend you to read the references in detail and correct them.   

 Additional minor revisions:

  1.  Page 2 line 17: “is” is lacking before “growing”

  2. Page 2 paragraph 1.3: consider moving the text between “Research is ongoing” and “BSI management in these settings” as this part seems more suitable for the “Discussion” than the “Introduction”. This allows you to simplify and shorten the introduction that is still well-written.

  3.  Page 4 paragraph 2.5: the link is not working. However, it should be provided as a reference, not like this

Author Response

First, we would like to thank all four reviewers for their constructive feedback. We have considered all suggestions, and strongly believe that we can now resubmit an improved manuscript. Below, detailed answers on all questions are given; the line numbers referred to are the line numbers in the track changes version.

Reviewer 4

This paper is very interesting and provides useful  additional data for bacterial detection prototypes. In general, the  paper is well written. “Materials and Methods” are easy to follow and  also provide a complete and clear description of the device. “Results”  are well presented, tables and figures are adequate. Lastly,  “Discussion” is properly argued. 

However, additional modifications can improve the work. 

Major modifications:

  1. The introduction could be shortened as some phrases are more suitable in “Discussion”

Thank you for the suggestion, we have considered this, but since the paragraph shapes the context for our development, we decided to leave it in the introduction, while focusing in the discussion on the turbidimeter itself.

  1. More importantly, the references are missing in the Introduction part of the manuscript.

Thank you for noticing, we have added the missing references.

  1. Another concern of mine is the fact that some references  that I decided to read (such as reference 9), do not have anything in  common with the text, but are cited. These unintentional and accidental  mistakes could have happened while writing, so I would recommend you to  read the references in detail and correct them.

We are very sorry about this problem with our reference manager that we didn’t notice. We have now checked all the references and corrected where needed.

 Additional minor revisions:

  1.  Page 2 line 17: “is” is lacking before “growing”

This correction was made.

  1. Page 2 paragraph 1.3: consider moving the text between  “Research is ongoing” and “BSI management in these settings” as this  part seems more suitable for the “Discussion” than the “Introduction”.  This allows you to simplify and shorten the introduction that is still  well-written.

Thank you for the suggestion, we have considered this, but since the paragraph shapes the context for our development, we decided to leave it in the introduction, while focusing in the discussion on the turbidimeter itself.

  1.  Page 4 paragraph 2.5: the link is not working. However, it should be provided as a reference, not like this

We have added the url to the reference list and have checked the correctness, by copy-pasting the link.

Round 2

Reviewer 3 Report

The authors have satisfactorily addressed the comments of the rveiewer.

Reviewer 4 Report

Manuscript modifications suggested by all reviewers improved this paper which was already interesting. This original work provides useful information for future research and possible clinical use.